# The validity and reliability of school-based fundamental movement skills screening to identify children with motor difficulties

**Lucy H. Eddy** [1,2]*, **Nick Preston**[2,3], **Shania Boom**[4], **Jessica Davison**[2,5], **Rob Brooks** [4], **Daniel D. Bingham**[4,6], **Mark Mon-Williams**[2,5,6], **Liam J. B. Hill**[2,5]

**1** Department of Psychology, University of Bradford, Bradford, United Kingdom, **2** Centre for Applied Education Research, Wolfson Centre for Applied Health Research, Bradford, United Kingdom, **3** Leeds Institute of Medical Research, University of Leeds, Leeds, United Kingdom, **4** Faculty of Health Studies, University of Bradford, Bradford, United Kingdom, **5** School of Psychology, University of Leeds, Leeds, United Kingdom, **6** Bradford Institute for Health Research, Bradford Royal Infirmary, Bradford, United Kingdom

* L.H.Eddy@bradford.ac.uk

**Data Availability Statement:** The data for this manuscript is available from the Open Science Framework via this link: https://osf.io/qr45t/?view_only=.

## Abstract

### Aim

Assess whether school-based teacher-led screening is effective at identifying children with motor difficulties.

### Methods

Teachers tested 217 children aged between 5 and 11 years old, after a one hour training session, using a freely available tool (FUNMOVES). Four classes ($n = 91$) were scored by both researchers and teachers to evaluate inter-rater reliability. Researchers assessed 22 children using the Movement Assessment Battery for Children (MABC-2; considered to be the 'gold standard' in Europe for use as part of the diagnostic process for Developmental Coordination Disorder) to assess concurrent and predictive validity.

### Results

Inter-rater reliability for all individual activities within FUNMOVES ranged from 0.85–0.97 (unweighted Kappa; with 95%CI ranging from 0.77–1). For total score this was lower ($\kappa = 0.76$, 95%CI = 0.68–0.84), however when incorporating linear weighting, this improved ($\kappa = 0.94$, 95%CI = 0.89–0.99). When evaluating FUNMOVES total score against the MABC-2 total score, the specificity (1, 95%CI = 0.63–1) and positive predictive value (1; 95%CI = 0.68–1) of FUNMOVES were high, whereas sensitivity (0.57, 95%CI = 0.29–0.82) and negative predictive values (0.57, 95%CI = 0.42–0.71) were moderate. Evaluating only MABC-2 subscales which are directly related to fundamental movement skills (Aiming & Catching, and Balance) improved these values to 0.89 (95%CI = 0.52–1) and 0.93 (95%CI = 0.67–0.99) respectively.

**Funding:** The work of the lead author (L.H. Eddy) was supported by an Economic and Social Research Council (ESRC) Postdoctoral Fellowship (ES/X006050/1). M. Mon-Williams was supported by a Fellowship from the Alan Turing Institute. The work was conducted within infrastructure provided by the Centre for Applied Education Research (funded by the Department for Education through the Bradford Opportunity Area) and ActEarly: a City Collaboratory approach to early promotion of good health and wellbeing funded by the Medical Research Council (grant reference MR/S037527/). L.J.B. Hill, M. Mon-Williams, N. Preston and D. D. Bingham's involvement was supported by the National Institute for Health Research Yorkshire and Humber ARC (reference: NIHR20016). The views expressed in this publication are those of the author(s) and not necessarily those of the National Institute for Health Research or the Departments of Health and Social Care or Education. The funders had no role in study design, data collection and analysis, decision to publish, or preparation of the manuscript.

**Competing interests:** The authors have declared that no competing interests exist.

## Interpretation

Teacher-led screening of fundamental movement skills (via FUNMOVES) is an effective method of identifying children with motor difficulties. Such universal screening in schools has the potential to identify movement difficulties and enable earlier intervention than the current norm.

## 1. Introduction

There is growing evidence that children both in the UK [1] and internationally [2, 3], are struggling to develop motor skills that were the norm for equivalently aged children from previous generations. There appears to be a particularly worrying downtrend in one specific group of motoric abilities known as fundamental movement skills (e.g. running, jumping, hopping, throwing, kicking and balance), which provide the building blocks for developing more complex movement patterns [4]. This is particularly concerning as motor difficulties in childhood are known to have a detrimental effect on education, health and wellbeing outcomes [5–7], which can ultimately impact social mobility [8, 9].

In the UK, there are multiple issues with the pathway to receive diagnosis and support for motor difficulties. Children need a diagnosis of Developmental Coordination Disorder or ICD equivalent (of which fundamental movement skills form part of the assessment criteria) to access support for these skills, but the healthcare system is complex and inequalities mean that parents/guardians of low socioeconomic position and ethnic minorities are less likely to access their family doctor for support [10]. Additionally, children with difficulties won't be eligible for a diagnosis of Developmental Coordination Disorder if they have late onset of 'symptoms', if their difficulties don't 'significantly and persistently' interfere with activities of daily living (e.g. dressing oneself) and academic achievement or productivity at school, or if they have an intellectual disability, visual impairment or neurological condition affecting movement [11]. Finally, there will be children that do not meet the criteria for this diagnosis but still have substantive functional motor difficulties and are therefore unable to access services. Considering all of these factors together, it is likely that many children with motor difficulties are not being identified and supported in a timely manner. This aligns with research which has shown international parental dissatisfaction for the support available for children with motor difficulties [12–15].

The impending 'cost of living' crisis is likely to have a disproportionate effect on disadvantaged families, further worsening inequality in these sections of society [16, 17]. It is therefore imperative that children with additional needs can be identified in a timely manner to ensure they do not fall further behind their peers. One way to achieve this could be to universally screen children for difficulties in education settings. Unfortunately, research has shown that pre-existing clinical tools for diagnosing Developmental Coordination Disorder, such as the Movement Assessment Battery for Children (MABC) [18], are not feasible for use in schools at scale as it is time consuming (1 hour per child), costly to purchase (~£1,000), and cannot be implemented by teachers, only trained clinicians [19, 20]https://www.zotero.org/google-docs/?KdchXa. In addition, less time consuming methodologies (such as proxy-report questionnaires) can be highly subjective and require knowledge about what constitutes a good level of motor development [21]. Research suggests that teachers often have low levels of understanding of fundamental movement skills [19] and thus it is likely these methods wouldn't accurately identify children who would benefit from additional support.

To tackle these issues, a school-based screening tool (FUNMOVES) was developed with Primary schools in Bradford [22]. FUNMOVES allows two members of teaching staff to assess the fundamental movement skills of a whole class (~30 pupils) in under an hour [22]. The content and structural validity of FUNMOVES has been demonstrated when it is delivered by trained teachers [22]. An important next step in establishing the feasibility of school-based screening is to test whether teachers are reliable in their assessments and check that these tools are valid at detecting children who would likely receive a diagnosis of Developmental Coordination Disorder if they were referred into the health system for more comprehensive assessment.

This research aimed to evaluate inter-rater reliability (teacher vs researcher scoring), as well as concurrent and predictive validity of FUNMOVES against the MABC-2 [18]. The MABC-2 is endorsed by the European Academy for Childhood Disability as the 'gold standard' for use as part of the diagnostic process for Developmental Coordination Disorder [23].

## 2. Materials and methods

### 2.1 Recruitment

One school was recruited using contacts within Born in Bradford, the Centre for Applied Education Research and the local Department for Education Opportunity Area, who have well-established links to schools within the Bradford District area. The school was approached in May 2022 by: (i) a poster emailed to the schools detailing the purpose and design of the study and (ii) a follow-up face to face meeting with a trained researcher. The school involved was located within the 10% most deprived neighbourhoods in England (Index of Multiple Deprivation (IMD) Decile 1).

As FUNMOVES is a whole class assessment (i.e. is group-based in nature), the Headteacher verbally consented to classes taking part during lesson time. Once the school had opted-in to the study, information sheets and written opt-out consent forms were sent to parents. All children in participating schools gave verbal assent on the day of testing. All children in years 1–6 (aged 5–11 years old) were invited to participate. Children with Special Educational Needs were not excluded. The only exclusion criteria were non-consent (parental), non-assent (child). Ethical approval for this study was granted by the University of Leeds School of Psychology Ethics Committee (reference: PSYC-85).

### 2.2 Participants

**2.2.1 Inter-rater reliability.**   A subsample of 119 children (spanning years 1–6) were scored by both researchers and teachers.

**2.2.2 Concurrent and predictive validity.**   The school was asked to identify children they believed had motor difficulties (based on teacher judgement), to undertake an additional researcher-led assessment of motor ability–the Movement Assessment Battery for Children (second edition) (MABC-2) [18]. Once data were collected, and linked to FUNMOVES scores, researchers only had access to data which contained participant IDs rather than names.

### 2.3 Materials

**2.3.1 FUNMOVES.**   Teaching staff received a one hour training session and a FUNMOVES manual. The manual comprised: (i) a definition of fundamental movement skills, and why they are important for childhood development; (ii) a list of materials necessary to implement FUNMOVES (25 beanbags, a tape measure and chalk or electrical tape); (iii) how to implement and score each activity; (iv) score sheets to record the ability levels within their

class. Score sheets required teachers to note demographic information for each child including their sex, and preferred hand. A full breakdown of the activities involved in FUNMOVES, and how it was developed can be seen in Eddy et al. (2021) [22].

**2.3.2 Concurrent and predictive validity.** A subsample of participants were also tested using the MABC-2. The MABC-2 was chosen for a number of reasons. Firstly, a recent systematic review [20] showed that there are low correlations between product- and process-oriented measures of fundamental movement skills. Additionally, research suggests that these two types of assessment measure different aspects of fundamental movement skills [24]. Thus, as FUN-MOVES is a product-oriented measure, the assessment chosen to evaluate concurrent validity also needed to be product-oriented (as is the case with the MABC-2). The review also found that of all product-oriented measures of fundamental movement skills, the MABC was the most comprehensively evaluated against the Consensus-based Standards for the Selection of health status Measurement Instruments (COSMIN) checklist [25]. Finally, the MABC has been endorsed in the European Academy for Childhood Disability clinical guidelines as the 'gold standard' motor function assessment tool for use as part of the diagnostic process for Developmental Coordination Disorder [23], and is therefore used in clinical settings in the United Kingdom. If children are to be referred for more comprehensive assessment based on their responses on FUNMOVES, it is important that FUNMOVES maps to clinically used tools to reduce the pressure on already over-stretched healthcare services [26].

### 2.4 Procedure

**2.4.1 FUNMOVES.** This study was observational in nature and took place in June 2022. Prior to testing, researchers hosted a training workshop for all teachers and teaching assistants in the school. The workshop explained the importance of measuring fundamental movement skills before teachers role-played in interactive sessions to practise leading and scoring all of the activities within FUNMOVES. Teachers were encouraged to ask questions throughout. They were also provided with contact details for the lead researcher to ensure any questions they had were answered prior to testing. At the end of training, each teacher was given score sheets and were asked to place their pupils in groups of five based on ability levels in Physical Education classes.

The FUNMOVES assessment is carried out within a five-by-five metre grid. A minimum of two members of teaching staff was required to assess each class. Teachers explained and demonstrated each activity to the whole class before children were scored. Participants were not permitted to practise, and all participants were tested on one activity before the next was introduced. Implementation fidelity was not assessed as this has was extensively evaluated throughout the development of the tool to ensure that the final iteration of FUNMOVES was easy for teachers to lead and score [22]. After testing, the school was given individual reports detailing how each pupil performed relative to the rest of their year group across all activities, calculated using percentile rank.

**2.4.2 Inter-rater reliability.** Two trained researchers scored children, in parallel to teachers, for four classes, thus ensuring that each year group was included (classes comprised two year groups i.e. Years 1 and 2, Years 3 and 4 and Years 5 and 6 were taught together).

**2.4.3 Concurrent and predictive validity.** Thirty children were identified by teachers as having poor motor skills and were referred for a researcher-led assessment of fundamental movement skills (MABC-2). Four researchers undertook this aspect of testing, three of which were trained by the fourth assessor (a qualified Occupational Therapist with experience of undertaking MABC assessments in a clinical capacity) in a half-day workshop prior to testing, where researchers practised demonstrating and scoring each activity. Participants were taken

out of standard lessons for approximately 40–60 minutes to complete all three subscales of the MABC—Manual Dexterity, Aiming and Catching and Balance. All MABC-2 assessments were conducted in a quiet room in the school, away from distractions.

## 2.5 Analysis

All analyses were conducted in R and R Studio (versions 4.2.1). Alpha level was 0.05 for all analyses.

**2.5.1 Inter-rater reliability.** Agreement between teachers and researchers (inter-rater reliability), was measured using Kappa statistics (both unweighted and linear weighted values were calculated).

**2.5.2 Concurrent and predictive validity.** Agreement between tests (concurrent validity) and predictive validity were measured using sensitivity, specificity, positive predictive value, negative predictive value, and accuracy. Definitions for each of these and how they were calculated can be found in Table 1.

Percentiles for FUNMOVES were calculated based on performance compared to the rest of the year group in the school. MABC percentiles were calculated using the standardised normative data for total score. For FUNMOVES a total score is derived from a sum of all activities. Information on calculating a total score for the MABC-2 can be found in the user manual [18]. FUNMOVES total score was also compared to the average percentile for MABC-2 subscales which contain fundamental movement skills (Aiming & Catching and Balance), removing manual dexterity, to generate a more directly comparable measure.

# 3. Results

## 3.1 Participants

No opt-out consent forms were returned to the school, so all children (present on days of testing) took part. A total of 217 children across school years 1–6 (ages 5–11 years) from a Primary School in Bradford participated in this research. Of this 217, a subsample of 119 children (one class per year group [1–6]) were scored by both researchers and teachers to evaluate inter-rater reliability. For concurrent and predictive validity, the school was asked to identify children that they believed had motor difficulties, out of the 217 recruited. Teachers identified 30 children, five from each year group. Of this subsample, 22 completed both the MABC and FUNMOVES, due to school absences (children who had full parental consent but were not present on days of testing), and thus were included in analyses. For demographic information about these samples, see Table 2. Nine teachers and nine teaching assistants scored the children using FUNMOVES.

## 3.2 Inter-rater reliability

Inter-rater reliability was 'almost perfect' for all individual activity scores [28]. The total score had 'substantial' levels of agreement [28], however when incorporating linear weighting this value improved substantially (see Table 3).

Difference scores were also calculated to evaluate how far from researcher scoring teachers were. For a full breakdown by activity see Table 3.

## 3.3 Concurrent and predictive validity

The proportion of true positives, true negatives, false positives and false negatives can be found in Table 4.

**Table 1. Concurrent and predictive validity analyses.**

| Construct | Definition | Operationalisation | Formula |
|---|---|---|---|
| Sensitivity | The ability of a test to correctly classify an individual as having the disease [27]. | The proportion of children that scored <15th percentile on FUNMOVES **and** the MABC-2. | TP/(TP +FN) |
| Specificity | The ability of a test to correctly classify an individual as not having the disease [27]. | The proportion of children that scored >15th percentile on FUNMOVES **and** the MABC-2. | TN/(TN +FP) |
| Positive predictive value (PPV) | The percentage of individuals with a positive test who actually have the disease [27]. | The probability that a child that scored <15th percentile on FUNMOVES also scored <15th percentile on the MABC-2. | TP/(TP +FP) |
| Negative predictive value (NPV) | The percentage of individuals with a negative test who do not have the disease [27]. | The probability that a child that scored >15th percentile on FUNMOVES also scored >15th percentile on the MABC-2. | TN/(TN +FN) |
| Accuracy | The percentage of children that are classified correctly | What is the percentage of children that FUNMOVES correctly classifies (in alignment with the MABC-2) as at risk of motor development (<15th percentile), and 'normal' (>15th percentile) motor development? | (TP+TN)/ Total |

*NB*: TP = True Positive, TN = True Negative, FP = False Positive, FN = False Negative

All six children with false negatives scored below the 15th percentile on the Manual Dexterity subscale (mean = 5.08, *SD* = 4.32, 95%CI = 1.62–8.54). None of these children scored below the 15th percentile on the Aiming and Catching subscale (mean = 36, *SD* = 22.27, 95% CI = 18.18–53.82). Five scored below the 15th percentile on the Balance subscale (mean = 7.52, *SD* = 5.37, 95%CI = 2.81–12.23). With Aiming and Catching and Balance subscale percentiles averaged (calculating a fundamental movement skill only total score; mean = 22.33, *SD* = 10.67, 95%CI = 13.79–30.87), only one false negative remained. There were no false positives (i.e. children who would be accidentally referred, and thus would waste resources) in any combination of scores.

To further assess the utility of FUNMOVES as a screening tool for both overall motor ability (MABC-2 Total Score) and fundamental movement skill ability (MABC-2 Aiming & Catching and Balance), sensitivity, specificity, positive predictive value, negative predictive value and accuracy were calculated (see Table 5).

## 4. Discussion

This research aimed to, for the first time, assess the inter-rater reliability, concurrent validity and predictive validity of FUNMOVES. The results show that teachers were able to score all activities within FUNMOVES in alignment with researchers after an hour of group training (unweighted Kappa > 0.85). This is a positive step forwards as many common motor skill assessment tools require trained professionals to deliver them, precluding their use by teaching staff [29]. To effectively screen children for fundamental movement skill difficulties, schools

**Table 2. Demographic information about the sample.**

| Demographic | Whole sample (n = 217) | Subsample used for inter-rater reliability (n = 119) | Subsample used for concurrent and predictive validity (n = 22) |
|---|---|---|---|
| Sex | | | |
| Male n (%) | 98 (45%) | 56 (47%) | 11 (50%) |
| Female n (%) | 119 (55%) | 63 (53%) | 11 (50%) |
| Mean age (SD) | 7.90 (1.66) | 8.69 (1.62) | 8.56 (1.34) |
| Special Educational Needs status n (%) | 37 (17%) | 21 (18%) | 8 (36%) |

**Table 3. Inter-rater reliability sensitivity of scoring.**

| Activity | Unweighted Kappa (95%CI) | Linear weighted Kappa (95%CI) | | Teacher score in relation to researcher scoring | | | | | | |
|---|---|---|---|---|---|---|---|---|---|---|
| | | | | -3 | -2 | -1 | 0 | 1 | 2 | 3 |
| Running | 0.85 (0.77–0.93) | 0.90 (0.84–0.95) | n | 0 | 1 | 2 | 106 | 10 | 0 | 0 |
| | | | % | 0 | 0.8 | 2 | 89 | 8 | 0 | 0 |
| Jumping | 0.96 (0.93–1) | 0.98 (0.95–1) | n | 0 | 0 | 2 | 116 | 1 | 0 | 0 |
| | | | % | 0 | 0 | 2 | 97 | 0.8 | 0 | 0 |
| Hopping | 0.95 (0.89–1) | 0.95 (0.90–1) | n | 0 | 0 | 1 | 115 | 2 | 1 | 0 |
| | | | % | 0 | 0 | 0.8 | 97 | 2 | 0.8 | 0 |
| Throwing (dominant) | 0.97 (0.93–1) | 0.98 (0.95–1) | n | 0 | 0 | 2 | 117 | 0 | 0 | 0 |
| | | | % | 0 | 0 | 2 | 98 | 0 | 0 | 0 |
| Throwing (non-dominant) | 0.97 (0.93–1) | 0.98 (0.95–1) | n | 0 | 0 | 1 | 117 | 1 | 0 | 0 |
| | | | % | 0 | 0 | 0.8 | 98 | 0.8 | 0 | 0 |
| Kicking | 0.95 (0.89–1) | 0.96 (0.91–1) | n | 0 | 0 | 1 | 115 | 3 | 0 | 0 |
| | | | % | 0 | 0 | 0.8 | 97 | 3 | 0 | 0 |
| Balance | 0.91 (0.85–0.98) | 0.94 (0.89–0.99) | n | 0 | 0 | 2 | 112 | 5 | 0 | 0 |
| | | | % | 0 | 0 | 2 | 94 | 4 | 0 | 0 |
| Total Score | 0.76 (0.68–0.84) | 0.94 (0.89–0.99) | n | 1 | 0 | 6 | 93 | 18 | 1 | 0 |
| | | | % | 0.8 | 0 | 5 | 78 | 15 | 0.8 | 0 |

*NB*:, CI = confidence interval

will need to be self-sufficient (with appropriate support in place) after training, and this would be made possible through the use of FUNMOVES.

It is important that FUNMOVES and the MABC-2 identify the same children as needing additional support. With healthcare services already stretched [26], it is vital that any children that are referred to these services do indeed need specialist healthcare support. Analyses revealed that the probability of FUNMOVES correctly classified all children that do not have motor difficulties (i.e. did not fall below the 15th percentile on the MABC-2; specificity = 1, 95%CI = 0.63–1). Additionally, FUNMOVES had a positive predictive value of 1 (95% CI = 0.68–1)such that the percentage of children who score below the 15th percentile that also scored below this threshold on the MABC-2 total score was high. Conversely, negative predictive value (0.57, 95%CI = 0.42–0.71) and sensitivity (0.57, 95%CI = 0.29–0.82) were moderate, meaning that some children who score above the 15th percentile on FUNMOVES may be classified as 'below average' by the MABC-2 total score. Additionally, accuracy of FUNMOVES for correctly classifying children was 73%. It is likely that part of the reason for this is that the MABC-2 total score also incorporates manual dexterity (fine motor) skills (e.g., placing pegs, drawing a trail, and threading beads). FUNMOVES focuses on fundamental movement (gross

**Table 4. True positives, true negatives, false positives and false negatives.**

| | True Positives | True Negatives | False Positives | False Negatives |
|---|---|---|---|---|
| *MABC-2 Total score* | 8 | 8 | 0 | 6 |
| *MABC-2 Aiming & Catching and Balance subscales (fundamental movement skills)* | 8 | 13 | 0 | 1 |

NB: True positive = child scored < 15th percentile on FUNMOVES and MABC-2, True negative (= child scored >15th percentile on FUNMOVES and MABC-2, False positive = child scored <15th percentile on FUNMOVES, but scored >15th percentile on MABC-2, and False negative = child scored > 15th percentile on FUNMOVES but <15th percentile on the MABC-2.

**Table 5. Concurrent and predictive validity for FUNMOVES.**

| Analysis | Against MABC-2 Total Score (95% CI) | Against MABC-2 Aiming & Catching and Balance (95% CI) |
|---|---|---|
| Sensitivity | 0.57 (0.29–0.82) | 0.89 (0.52–1) |
| Specificity | 1 (0.63–1) | 1 (0.75–1) |
| Positive predictive value | 1 (0.68–1) | 1 (0.68–1) |
| Negative predictive value | 0.57 (0.42–0.71) | 0.93 (0.67–0.99) |
| Accuracy | 0.73 (0.50–0.89) | 0.95 (0.77–1) |

NB: CI = confidence interval; see Table 1 for definitions

motor) skills, which are a separate, distinct group of motor abilities. This is reflected in the fact that all of the false negatives scored below the 15th percentile on the manual dexterity subscale.

Research has suggested that teachers are well placed to identify children that have difficulties with fine motor skills, such as handwriting [30], which is perhaps expected given that this is often the method used to assess a child's academic ability in school. Contrastingly, research has demonstrated a poor level of understanding amongst teachers about fundamental movement skills [19]. When looking at the two subscales of the MABC-2 which directly link to fundamental movement skills, the classification analyses were all strong (sensitivity = 0.89, specificity = 1, positive predictive value = 1, negative predictive value = 0.93 and accuracy = 95%). These preliminary results demonstrate that FUNMOVES may be an effective method of screening children for fundamental movement skill difficulties.

It is, however, important to consider that the sample size for these aspects of validity were small ($n$ = 22). This limits the conclusions that can be drawn. Similarly, teachers were only asked to refer children they thought had motor skill difficulties for researcher-led assessment on the MABC-2, which could impact the generalisability of the results (along with convenience sampling)–however children within this sample did score within 'normal' ranges, showing a range of abilities. In addition to this, the children that were assessed on FUNMOVES received percentile scores based on their performance compared to the rest of the children in their year group. This does not necessarily represent their performance compared to a broader, nationally representative sample. It would be beneficial to formulate a normative database to understand fundamental movement skill ability levels beyond Bradford. Once a representative normative database has been formed, it would be beneficial to conduct this research on a larger scale. Nevertheless, the current study shows that such a programme of work is well justified.

FUNMOVES raw scores were converted into Percentile Ranks relative to the age of the children in the study. This was done to align with the MABC-2, and enable easy comparison of children struggling with motor development using the <15th percentile cut off. It is, however important to note that there are a number of problems with use of these standardisation metrics which need to be considered. Firstly, it is likely that standardised samples do not include percentile ranks for each raw score that is achievable on the test, and secondly, extreme scores can distort the assignment of percentile ranks [31]. When developing a normative database for FUNMOVES it will be important to consider more continuous solutions to avoid these pitfalls to ensure that the correct children are identified as having difficulties [31]. The results did show though that FUNMOVES is a strong predictor of fundamental movement skill difficulties, as assessed by the MABC-2. By embedding FUNMOVES into school Physical Education lessons, and using its insights alongside teacher knowledge about their pupils' classroom skills (e.g. handwriting and using scissors), there is the potential to reduce inequalities and improve

access to support (provided there is sufficient communication and collaboration between education and health to facilitate quicker referrals based on FUNMOVES scores). It is, however, important to note that the MABC-2 isn't the only metric used to detect Developmental Coordination Disorder [23], so children who score poorly on FUNMOVES may not always be eligible for a diagnosis. It is crucial that children waiting for diagnoses (considering the lengthy waiting time for assessment) and those who have difficulties but do not meet the criteria for diagnosis, that would also benefit from additional support to reach their full potential, are able to access it.

Previous research has shown that school-based assessment of motor skill difficulties (through placing Occupational Therapists within the classroom environment) can be effective at reducing time to diagnosis [32]. This research has shown that FUNMOVES may prove to be an effective way to triage access to services using a universal approach. Moreover, if paired with school-based intervention, FUNMOVES could help children receive more timely support as part of a needs-based approach, without the need for a diagnosis (thus supporting those who do not meet clinical thresholds for Developmental Coordination Disorder but still have substantive functional difficulties). This shift towards 'participation' in alignment with the international classification of functioning, disability and health framework developed by the World Health Organisation [33] will likely have broad-ranging benefits for population level education, health, wellbeing and life chances [5–7].

## 5. Conclusions

Screening children's fundamental movement skills in schools using FUNMOVES has the potential to expedite access to further assessment and intervention through facilitating increased communication and collaboration between healthcare, families and education. FUNMOVES therefore offers a unique opportunity to use a robustly developed assessment with strong validity and reliability to universally screen ability levels in schools.

## Author Contributions

**Conceptualization:** Lucy H. Eddy, Daniel D. Bingham, Mark Mon-Williams, Liam J. B. Hill.

**Data curation:** Lucy H. Eddy, Shania Boom, Jessica Davison, Rob Brooks, Daniel D. Bingham.

**Formal analysis:** Lucy H. Eddy, Liam J. B. Hill.

**Funding acquisition:** Lucy H. Eddy, Liam J. B. Hill.

**Investigation:** Lucy H. Eddy, Shania Boom, Jessica Davison, Rob Brooks, Liam J. B. Hill.

**Methodology:** Lucy H. Eddy, Nick Preston, Mark Mon-Williams, Liam J. B. Hill.

**Project administration:** Lucy H. Eddy, Shania Boom, Jessica Davison, Rob Brooks.

**Resources:** Lucy H. Eddy, Mark Mon-Williams.

**Supervision:** Nick Preston, Daniel D. Bingham, Mark Mon-Williams, Liam J. B. Hill.

**Writing – original draft:** Lucy H. Eddy.

**Writing – review & editing:** Lucy H. Eddy, Nick Preston, Shania Boom, Jessica Davison, Rob Brooks, Daniel D. Bingham, Mark Mon-Williams, Liam J. B. Hill.

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
