## [Decision Letter · Decision Letter 0]

3 Jan 2023

PONE-D-22-29655

The validity and reliability of school-based fundamental movement skills screening to identify children with motor difficulties

PLOS ONE

Dear Dr. Eddy,

Thank you for submitting your manuscript to PLOS ONE. After careful consideration, we feel that it has merit but does not fully meet PLOS ONE’s publication criteria as it currently stands. Therefore, we invite you to submit a revised version of the manuscript that addresses the points raised during the review process.

We look forward to receiving your revised manuscript.

Kind regards,

Gorica Maric

Academic Editor

PLOS ONE

Journal Requirements:

   "The work of the lead author (L.H. Eddy) was supported by an ESRC White Rose Doctoral Postdoctoral Research Fellowship. M. Mon-Williams was supported by a Fellowship from the Alan Turing Institute. The work was conducted within infrastructure provided by the Centre for Applied Education Research (funded by the Department for Education through the Bradford Opportunity Area) and ActEarly: a City Collaboratory approach to early promotion of good health and wellbeing funded by the Medical Research Council (grant reference MR/S037527/). L.J.B. Hill, M. Mon-Williams, N. Preston and D. D. Bingham’s involvement was supported by the National Institute for Health Research Yorkshire and Humber ARC (reference: NIHR20016). The views expressed in this publication are those of the author(s) and not necessarily those of the National Institute for Health Research or the Departments of Health and Social Care or Education. The funders had no role in study design, data collection and analysis, decision to publish, or preparation of the manuscript. "

 "The work of the lead author (L.H. Eddy) was supported by an ESRC White Rose Doctoral Postdoctoral Research Fellowship. M. Mon-Williams was supported by a Fellowship from the Alan Turing Institute. The work was conducted within infrastructure provided by the Centre for Applied Education Research (funded by the Department for Education through the Bradford Opportunity Area) and ActEarly: a City Collaboratory approach to early promotion of good health and wellbeing funded by the Medical Research Council (grant reference MR/S037527/). L.J.B. Hill, M. Mon-Williams, N. Preston and D. D. Bingham’s involvement was supported by the National Institute for Health Research Yorkshire and Humber ARC (reference: NIHR20016). The views expressed in this publication are those of the author(s) and not necessarily those of the National Institute for Health Research or the Departments of Health and Social Care or Education. The funders had no role in study design, data collection and analysis, decision to publish, or preparation of the manuscript. "

Additional Editor Comments :

Dear authors, please revise your manuscript according to the referees’ comments and upload the revised file. You should provide more details on sampling procedure and criteria for selection. A database used should be attached as well. I have to agree with Reviewer number 1 that your hypotheses at the end of Introduction section seem identical to the results you present in your work. Also, authors should reevaluate appropriateness of statistical tests used in the study as indicated by both reviewers. Furthermore, we recommend the authors to revise terminology used throughout the manuscript. At the end of your revision, please, tailor study limitations accordingly.

Reviewers' comments:

Reviewer's Responses to Questions

**Comments to the Author**

1. Is the manuscript technically sound, and do the data support the conclusions?

Reviewer #1: Partly

Reviewer #2: Yes

2. Has the statistical analysis been performed appropriately and rigorously? 

Reviewer #1: No

Reviewer #2: Yes

3. Have the authors made all data underlying the findings in their manuscript fully available?

Reviewer #1: Yes

Reviewer #2: Yes

4. Is the manuscript presented in an intelligible fashion and written in standard English?

Reviewer #1: Yes

Reviewer #2: Yes

5. Review Comments to the Author

Reviewer #1: This article reports a validation of a screening tool for detecting motor difficulties in children. While the article is well-written, this reviewer has major concerns about the form, the methodology, and the reporting.

General comments

- The authors often report statements that are not substantiated by the references they provide. I would encourage the authors to not report personal opinions and focus on the research question assessed in the article.

- Do not use unnecessary acronyms such as FMS, EACD, COSMIN, P.E., SEND. Please spell these acronyms out throughout the manuscript to improve readability.

- Details about the sampling, inclusion and exclusion criteria are insufficient.

- I believe the choice of the statistical methods not to be appropriate, and lacking a sound rationale for the purpose of validating a screening tool.

Abstract

-P2, lines 22 & 27. Since this is a validation study, the term “Explore” should be replaced with “Assess”.

-P2, line 35-36. Please remove this sentence “Universal screening of motor skills could help to reduce healthcare inequalities and improve life chances for children” from the end of the abstract. This statement is not substantiated from your data and is outside the scope of this validation study.

Introduction

-P3, line 45. Please provide a reference for this statement “can ultimately impact social mobility.”

-P3, lines 55-56. Please remove this specification “(as if another disorder decreases the impact of motor deficits)” unless the authors can point the readers towards a critical perspective article on this issue.

-P3, lines 56-58. Please provide a reference for the statement “All of these factors combined mean that children with motor difficulties are not having their needs identified (most prominently children of ethnic minorities from deprived areas)”

-The first part of the introduction is generally too long. While it is important to stress the importance of the screening tool, I think this part can be shortened without impacting the focus of the work and without harming readability

-P4, lines 75-76. Please provide a reference for “FUNMOVES allows two members of teaching staff to 76 assess the FMS of a whole class (~30 pupils) in under an hour.”

-Was the study pre-registered? The hypotheses at the end of the introduction seem very specific and tailored to the results obtained. Either the authors provide a link to a pre-registration or I think this section does not help in reading the article.

Methods

-P5, lines 103-104. The authors report that the first school to respond favourably was included in this study. It is unclear what happened to the other schools. Is other data available from the other schools?

-P6, lines 112-114. Please provide more details on how the sample size needed was estimated

-P8, line 176. Specify which version of R and RStudio was used for the analysis.

-The authors use ICC to measure the agreement between MABC-2 and FUNMOVES, and between teachers and researchers. I do not understand the rationale for this choice. The MABC-2 has been defined by the authors as the gold standard, and a comparison between a test and a gold standard should be based on Accuracy, Sensitivity, and Specificity. On the other hand, agreement between raters is generally evaluated using Kappa statistics. It is also possible to assign weights to different types of errors. If instead the issue is that the tests provide numeric results, the assessment should focus on metrics such as RMSE or graphical visualization through a scatterplot. Please either consider more appropriate metrics or provide a sound rationale for your choice.

-P9, lines 184-185. “Logistic regression was used to establish whether FUNMOVES and the MABC-2 identified the same children as having poor FMS.” Why is this needed? Was the logistic regression used to adjust for some variables? See the point above.

-I understand that the use of percentiles by year group to define individuals with a condition is customary in your field. But several studies have shown issues with standardization and correction in other fields (eg. measurement of lung functions, measurement of hearing functions, measurement of cognitive decline). Please be aware of these issues.

-P9, Table 1. The metrics described in Table 1 are known to the broad audience. Please, remove the table and describe in the text the cut-offs used.

Results

-Please provide a flow-chart indicating the number of contacted students and the number of students excluded by different reasons at different stages of the sampling procedure. This is a general recommendation for all observational studies (eg. STROBE guidelines). Information about how the authors arrived at the final sample is insufficient and does not allow judgements about representativeness.

-P10, line 202-204. This sentence belongs to the methods section or the discussion.

-P10, Table 2. please be consistent in your terminology: the term “sex” was used in the manuscript.

-P10, Table 2. report the number of individuals by sex along with the percentage (not the ratio).

-The MABC-2 was only assessed for students judged as having motor difficulties by their teachers. Therefore, the assessment of accuracy has limited value. This should be emphasized in the Discussion.

-Please, report confidence intervals for all the reported metrics (Sensitivity, Specificity, PPV, NPV, Accuracy). The sample size is very low and this should be reflected in a measure of variability. This assessment should, in my opinion, be the main point of the paper together with a Kappa Statistics about the raters agreement (see points above).

Discussion

-P13, line 247. This is not a “psychometric” test

-Correlation is not a measure of agreement. It measures linear relationships.

-P15, line 291-296. The use of norms by demographic characteristics has been shown to be problematic (see point above) and I do not think that your work “show that such programme of work is well justified”.

-P16, line 324-326. The final sentence in the conclusion is completely outside the scope of the article and should be removed.

Reviewer #2: The manuscript: The validity and reliability of school-based fundamental movement skills screening to identify children with motor difficulties is well written with detailed descriptions of methodology and results. Numerous advanced statistical methods have been applied. The Table are quite detailed. Less than 50% of references are older than 5 years. Both the topic of the study as well as the findings are interesting. However, I have some comments about the manuscript, as described below.

1. A database needs to be attached.

2. There is no total score and description for FUNMOVES and MABC-2

3. It is unclear how logistic regression was used? What is the outcome in logistic regression.

4. State the applied statistical tests in the methodology

5. It is not adequate to report the mean and SD when the coefficient of variation is >50%

6. The paper does not state the level of significance.

7. Uniformly report p-values

8. Report ICC with 95% CI, without F and p-value

9. Add leading zeros to numeric values before of decimal places.

6. PLOS authors have the option to publish the peer review history of their article (what does this mean?). If published, this will include your full peer review and any attached files.

Reviewer #1: No

Reviewer #2: No

---

## [Author Response · Author response to Decision Letter 0]

30 Mar 2023

Editor Comments 

We have updated the manuscript to reflect PLOS ONE formatting 

2. Remove funding information from acknowledgements. Include amended funding statement within cover letter. Specify where the minimal dataset is held in the cover letter

This has now been removed from the acknowledgements section and both of these things are now included in the cover letter. 

You should provide more details on sampling procedure and criteria for selection

This information has now been clarified in the methods section (see pages 5 and 6, lines 109 - 125).

3. I have to agree with Reviewer number 1 that your hypotheses at the end of the Introduction section seem identical to the results you present in your work.

These hypotheses were not pre-registered so have been removed.

4. Also, authors should reevaluate appropriateness of statistical tests used in the study as indicated by both reviewers. Furthermore, we recommend the authors to revise terminology used throughout the manuscript

We have taken on board Reviewer one’s comments and kept in the classification analyses, but removed intraclass correlations (replaced with Kappa for inter-rater reliability) and logistic regression from the manuscript.

5. At the end of your revision, please, tailor study limitations accordingly

The discussion has been modified to include limitations highlighted by the reviewers (see pages 16 & 17 , lines 332-335, 345-354). 

Reviewer 1 

General comments

1. The authors often report statements that are not substantiated by the references they provide. I would encourage the authors to not report personal opinions and focus on the research question assessed in the article.

We have reframed the introduction and discussion to ensure that the findings of the papers cited are reported with more clarity, and included further references to support statements made (see pages 3-5 and 14-18).

2. Do not use unnecessary acronyms such as FMS, EACD, COSMIN, P.E., SEND. Please spell these acronyms out throughout the manuscript to improve readability.

All acronyms (barring the Movement Assessment Battery for Children, as this is a particularly well known acronym within this research area) have now been removed from the manuscript.

3. Details about the sampling, inclusion and exclusion criteria are insufficient.

 I believe the choice of the statistical methods not to be appropriate, and lacking a sound rationale for the purpose of validating a screening tool.

Inclusion and exclusion criteria have now been clarified, and a flow diagram for sampling has now been included (as mentioned in a later comment). See pages 5&6 116 - 125. These analyses were chosen based on their frequency of use within the field for similar research questions. We appreciate your comments around more appropriate analyses, and have consequently removed the intraclass correlation results and logistic regression results from this paper. Instead, we focused on classification analyses and kappa statistics, as suggested in your comments below. 

Abstract 

1. P2, lines 22 & 27. Since this is a validation study, the term “Explore” should be replaced with “Assess”.

Thank you for this helpful clarification of the most appropriate terminology - this has now been changed throughout the manuscript. 

2. P2, line 35-36. Please remove this sentence “Universal screening of motor skills could help to reduce healthcare inequalities and improve life chances for children” from the end of the abstract. This statement is not substantiated from your data and is outside the scope of this validation study.

On reflection, we agree that this is beyond the scope of the paper and this has now been removed.

Introduction

1. P3, line 45. Please provide a reference for this statement “can ultimately impact social mobility.”

References have now been included in support of this statement (see page 3 line 49).

2. P3, lines 55-56. Please remove this specification “(as if another disorder decreases the impact of motor deficits)” unless the authors can point the readers towards a critical perspective article on this issue.

This has now been removed.

3. P3, lines 56-58. Please provide a reference for the statement “All of these factors combined mean that children with motor difficulties are not having their needs identified (most prominently children of ethnic minorities from deprived areas)”

This paragraph has been revised in alignment with the comment below.

4. The first part of the introduction is generally too long. While it is important to stress the importance of the screening tool, I think this part can be shortened without impacting the focus of the work and without harming readability

We agree there was room to be more succinct in this paragraph, whilst still maintaining key messages. We have now edited this section to reflect this request (see pages 3-4). 

5. P4, lines 75-76. Please provide a reference for “FUNMOVES allows two members of teaching staff to 76 assess the FMS of a whole class (~30 pupils) in under an hour.”

Apologies that it was not clear - the reference in the next sentence was to cover both. I have referenced this paper twice now for clarification.

6. Was the study pre-registered? The hypotheses at the end of the introduction seem very specific and tailored to the results obtained. Either the authors provide a link to a pre-registration or I think this section does not help in reading the article.

These hypotheses were not pre-registered, so have been removed.

Methods

1. P5, lines 103-104. The authors report that the first school to respond favourably was included in this study. It is unclear what happened to the other schools. Is other data available from the other schools?

There are no other schools involved in this study. This was a function of the timescale within which the study had to be run, due to funding limitations. This section has been reworded for clarification. See page 5 lines 109-115.

2. P6, lines 112-114. Please provide more details on how the sample size needed was estimated

The authors have reviewed this and feel that sufficient details have been included in this section. We are happy to be guided by the reviewer if they believe further information is necessary. 

3. P8, line 176. Specify which version of R and RStudio was used for the analysis.

This has now been included (see page 9 line 199).

4. The authors use ICC to measure the agreement between MABC-2 and FUNMOVES, and between teachers and researchers. I do not understand the rationale for this choice. The MABC-2 has been defined by the authors as the gold standard, and a comparison between a test and a gold standard should be based on Accuracy, Sensitivity, and Specificity. On the other hand, agreement between raters is generally evaluated using Kappa statistics. It is also possible to assign weights to different types of errors. If instead the issue is that the tests provide numeric results, the assessment should focus on metrics such as RMSE or graphical visualization through a scatterplot. Please either consider more appropriate metrics or provide a sound rationale for your choice.

We appreciate your comments and your suggestions for more appropriate analyses. In response we have modified the analysis in line with your recommendations (see response to general comment 3), see page 9, lines 201-220.

5. P9, lines 184-185. “Logistic regression was used to establish whether FUNMOVES and the MABC-2 identified the same children as having poor FMS.” Why is this needed? Was the logistic regression used to adjust for some variables? See the point above.

We thank the reviewer for highlighting the redundancy of this analysis. We have now removed the logistic regression analysis to improve clarity and readability.

6. I understand that the use of percentiles by year group to define individuals with a condition is customary in your field. But several studies have shown issues with standardization and correction in other fields (eg. measurement of lung functions, measurement of hearing functions, measurement of cognitive decline). Please be aware of these issues. 

We thank the reviewer for bringing this to our attention. We have now included this as a discussion point (see page 17, lines 345-354).

7. P9, Table 1. The metrics described in Table 1 are known to the broad audience. Please, remove the table and describe in the text the cut-offs used.

These are not regularly used metrics within our field, and thus, we deem it necessary to include this table within the paper. 

Results

1. Please provide a flow-chart indicating the number of contacted students and the number of students excluded by different reasons at different stages of the sampling procedure. This is a general recommendation for all observational studies (eg. STROBE guidelines). Information about how the authors arrived at the final sample is insufficient and does not allow judgements about representativeness.

There were no exclusions in the data. No non-consents were returned to the school so all children that were present on the days of testing took part. This has now been clarified (see page 10, lines 225-235). As there were no exclusions a flow chart was deemed excessive. It would only illustrate the number of children on the school roll, as all were invited to participate, flowing to a second box showing the number of children who did participate. 

2. P10, line 202-204. This sentence belongs to the methods section or the discussion.

We agree this gives context to the demographic of the school, rather than it being a result. This has now been moved to the methods section. See lines 97-99.

3. P10, Table 2. Please be consistent in your terminology: the term “sex” was used in the manuscript.

Apologies for the oversight, Table 2 has now been amended.

4. P10, Table 2. report the number of individuals by sex along with the percentage (not the ratio).

We thank the reviewer for recognising the difficulties that may occur when interpreting this table - we have now included n and percentage (where appropriate) - see page 11.

5. The MABC-2 was only assessed for students judged as having motor difficulties by their teachers. Therefore, the assessment of accuracy has limited value. This should be emphasized in the Discussion.

We appreciate the limitations with this methodology, however it was necessary due to time and resource limitations. We’d also note that the sample referred by teachers also included children that did not have motor skill difficulties (aligning with literature showing that teachers have a poor understanding of motor development). We have acknowledged this in the discussion (see page 16, lines 332-335).

6. Please, report confidence intervals for all the reported metrics (Sensitivity, Specificity, PPV, NPV, Accuracy). The sample size is very low and this should be reflected in a measure of variability. This assessment should, in my opinion, be the main point of the paper together with a Kappa Statistics about the raters agreement (see points above).

We agree that a measure of variability should be included, we have reported 95% confidence intervals in Table 5 (page 14). 

Discussion 

1. P13, line 247. This is not a “psychometric” test

Correlation is not a measure of agreement. It measures linear relationships.

Correlational analyses have now been removed from the paper and this wording has been amended (see page 14, lines 285-287).

2. P15, line 291-296. The use of norms by demographic characteristics has been shown to be problematic (see point above) and I do not think that your work “show that such programme of work is well justified”.

We appreciate the reviewer's opinion. However positive results in a preliminary study such as those presented in this manuscript (where the limitations of this are made explicit) do justify further exploration, in larger sample sizes. Regardless of whether percentile ranks are appropriate, we would argue that having more data (i.e. information on ability levels) will only increase sensitivity when comparing ability levels. We have included the limitations of using percentile ranks in the discussion (see page 17 lines 345-354).

3. P16, line 324-326. The final sentence in the conclusion is completely outside the scope of the article and should be removed.

This has now been removed.

Reviewer 2 

1. A database needs to be attached.

Information on where to access the database has now been given to PLOS ONE for inclusion in the manuscript.

2. There is no total score and description for FUNMOVES and MABC-2

This information has now been included on page 9, lines 217-219.

3. It is unclear how logistic regression was used? What is the outcome in logistic regression.

This analysis has since been removed, based on the recommendation of Reviewer 1. 

4. State the applied statistical tests in the methodology

This information can be found on page 9, lines 199 - 220.

5. It is not adequate to report the mean and SD when the coefficient of variation is >50%

We thank the reviewer for highlighting our omission. 95% confidence intervals have now been included (see Table 13, pages 11 and 12).

6. The paper does not state the level of significance.

This has now been included in the paper - see page 9, lines 199-200.

7. Uniformly report p-values

We have reported all p values as their true value, unless it is smaller than 0.001. This is common practice in our field.

8. Report ICC with 95% CI, without F and p-value

This analysis has now been removed based on the recommendation of Reviewer 1.

9. Add leading zeros to numeric values before of decimal places.

We have now added leading 0s to all decimal places throughout the manuscript.

---

## [Decision Letter · Decision Letter 1]

3 May 2023

PONE-D-22-29655R1The validity and reliability of school-based fundamental movement skills screening to identify children with motor difficultiesPLOS ONE

Dear Dr. Eddy,

Thank you for submitting your manuscript to PLOS ONE. After careful consideration, we feel that it has merit but does not fully meet PLOS ONE’s publication criteria as it currently stands. Therefore, we invite you to submit a revised version of the manuscript that addresses the points raised during the review process.

ACADEMIC EDITOR: Dear authors,  Thank you for implementing changes suggested by reviewers, you have now significantly improved your manuscript. However, there are still some methodological and terminological issues that require further evaluation before consideration of your text for the publication. Therefore, please, refer to Reviewer's comments and send us back re-revised version.

We look forward to receiving your revised manuscript.

Kind regards,

Gorica Maric

Academic Editor

PLOS ONE

Reviewers' comments:

Reviewer's Responses to Questions

**Comments to the Author**

1. If the authors have adequately addressed your comments raised in a previous round of review and you feel that this manuscript is now acceptable for publication, you may indicate that here to bypass the “Comments to the Author” section, enter your conflict of interest statement in the “Confidential to Editor” section, and submit your "Accept" recommendation.

Reviewer #1: (No Response)

Reviewer #2: All comments have been addressed

2. Is the manuscript technically sound, and do the data support the conclusions?

Reviewer #1: Partly

Reviewer #2: Yes

3. Has the statistical analysis been performed appropriately and rigorously? 

Reviewer #1: No

Reviewer #2: Yes

4. Have the authors made all data underlying the findings in their manuscript fully available?

Reviewer #1: Yes

Reviewer #2: Yes

5. Is the manuscript presented in an intelligible fashion and written in standard English?

Reviewer #1: Yes

Reviewer #2: Yes

6. Review Comments to the Author

Reviewer #1: The manuscript has undoubtedly improved. However, I still have major concerns regarding terminological and methodological elements.

Abstract

- The separation between inter-rater reliability (Kappa of researcher vs teachers ratings) and the predictive validity (FUNMOVES vs MABC-2 in the subgroup) is now clear in the main text. In the abstract however, this sentence is confusing: “Four classes (n=91) were scored by both researchers and teachers to evaluate inter-rater reliability against ‘gold standard’ scoring.” Please delete “against gold standard scoring”. Add in a sentence explaining that the MABC-2 is considered the gold standard in the predictive validity assessment.

- Report 95% confidence intervals for all metrics in the abstract.

- Use proper language when describing the results. If the authors are describing inter-rater agreement they should not call it “accuracy”, which is a term that indicates that the classification was correct. In this paper, the authors are only assessing if the classifications from different raters were in agreement.

- I am confused by this sentence in the abstract “The Specificity and positive predictive value of FUNMOVES was high (1). Sensitivity and negative predictive values were lower for MABC-2 total score (0.57)”. MABC-2 has been defined as the gold standard. Therefore, by definition, the sensitivity and specificity of the gold standard are both 1.00. The authors should report the metrics (sensitivity, specificity, PPV, NPV) for the FUNMOVES tool in identifying subjects classified with or without the condition (as defined by the MABC-2).

Introduction

- Introduction is much better now.

- Page 3, line 56. Two periods.

Materials and Methods

- The section 2.2.1 as it is, is not useful. Without specifying which test was considered and which parameters were used, those numbers have no meaning. I understand that the sample size calculation may have been conducted in the design phase when the planned statistical analyses were different from the ones conducted. Either provide a complete description of the rationale for the sample size calculation, or remove the text.

- Please modify Table 1 according to the following suggestions:

o Rename the column “Description” as “Definition”

o The definitions reported for sensitivity and specificity are incorrect. Please provide formal definitions. The text as it is, is very misleading. For example: 1) sensitivity is not “ability”, it is a probability. 2) sensitivity is not the probability of the test to correctly identify those who have motor difficulties…it’s the probability of being classified as having motor difficulties by the test GIVEN that the child actually has motor difficulties

o Add formal definition also for PPV and NPV

o Classification rate is not a metric. The metric described by the authors is called “Accuracy”. The definition of Accuracy should not be formulated as a question. Please add a formal definition.

o Add a column, called “Operationalization”. In this column, explain how you calculated the quantities for each metric, exactly as you had done for PPV and NPV in the column currently called “Description”.

o Remove the “(n=22)”. This is not the appropriate table to report sample sizes.

o Change the title of the table in accordance with the name given to the analyses in the text

Results

- Sample sizes are much more clear now.

- Please make the following edits to Table 2:

- Rename “inter-rater reliability” with “Subsample used for inter-rater reliability analysis”. Following the same rationale, also rename “concurrent and predictive validity”.

- In the first column, below “Sex”, there should be two cells. One cell with “Male, n(%)” and one with “Female, n(%)”. This way, you do not have to repeat in the other three columns “Male: n=” and “Female:=”. This is usually how sex is reported in Tables.

- Section 3.2 Inter-rater reliability. There cannot be a gold standard of inter-rater reliability by definition. Please avoid the use of this term when referring to the inter-rater reliability assessment, as it is incompatible with the chosen statistical methods.

- Table 3. Provide confidence intervals for the linear weighted Kappa. Please also delete the mention to “p<.001” in the caption, as no p-value is reported in the table.

- Table 4 should be a cross-table with the classification of the FUNMOVES and the one of MABC-2 total score.

-The concurrent and predictive validity has not been defined as a sensitivity analysis in the methods. Please modify the language.

- The MABC-2 Aiming and Catching and Balance subscales (FMS) has not been mentioned at all in the methods and it is unclear where it comes from.

Discussion

- When commenting on the results from the predictive validity analysis, please use correct definitions of the metrics (sensitivity, specificity, PPV, NPV). Do not oversimplify their meaning.

- The claim “the classification analyses were all strong” is unclear and is not scientific language. (1) Classification analysis is not a recognized term to refer to predictive validity analysis. (2) The phrase “all strong” is nondescript and I therefore don’t know what it means.

- Page 14, line 278. Should say “could impact the generalizability of results”. Please use appropriate terms for scientific concepts.

- Page 14, line 283 to page 15 line 288. The fact that demographic factors affect the condition (fundamental movement skill ability levels) does not justify the correction by demographic factors for the test. It has been mathematically proven that correction for demographic factors can reduce the accuracy of tests under certain causal scenarios (this has been shown extensively for cognitive screening tests, for example). It is not “essential to formulate a normative database” until it is shown that the correction for demographic factors improves the metric of interest. This has not been demonstrated in this paper and therefore, it is not true that this “current study shows that such a programme of work is well justified”.

- In the next paragraph of the discussion, the authors continue to refer to the need for a normative database.

- Line 326. “Psychometric” refers to psychological tests.

Reviewer #2: (No Response)

7. PLOS authors have the option to publish the peer review history of their article (what does this mean?). If published, this will include your full peer review and any attached files.

Reviewer #1: No

Reviewer #2: No

---

## [Author Response · Author response to Decision Letter 1]

15 May 2023

Abstract

 1. The separation between inter-rater reliability (Kappa of researcher vs teachers ratings) and the predictive validity (FUNMOVES vs MABC-2 in the subgroup) is now clear in the main text. In the abstract however, this sentence is confusing: “Four classes (n=91) were scored by both researchers and teachers to evaluate inter-rater reliability against ‘gold standard’ scoring.” Please delete “against gold standard scoring”. Add in a sentence explaining that the MABC-2 is considered the gold standard in the predictive validity assessment.

We agree this could be confusing, given the ‘gold standard’ terminology used for the MABC. We have now removed this and added in a sentence justifying the use of the MABC for concurrent and predictive validity analyses (see page 2, line 26). 

 2. Report 95% confidence intervals for all metrics in the abstract.

95% confidence intervals have been included throughout the abstract. See page 2 lines 29-40.

 3. Use proper language when describing the results. If the authors are describing inter-rater agreement they should not call it “accuracy”, which is a term that indicates that the classification was correct. In this paper, the authors are only assessing if the classifications from different raters were in agreement.

We agree that this could be confusing, given the change of terminology from classification rates to accuracy. This has now been modified(see page 2, lines 28-29). 

 4. I am confused by this sentence in the abstract “The Specificity and positive predictive value of FUNMOVES was high (1). Sensitivity and negative predictive values were lower for MABC-2 total score (0.57)”. MABC-2 has been defined as the gold standard. Therefore, by definition, the sensitivity and specificity of the gold standard are both 1.00. The authors should report the metrics (sensitivity, specificity, PPV, NPV) for the FUNMOVES tool in identifying subjects classified with or without the condition (as defined by the MABC-2).

The sensitivity, specificity, PPV, NPV for MABC subscales that are directly comparable to FUNMOVES have been clarified (see page 2, lines 36-39). 

Introduction

 5. Introduction is much better now. Page 3, line 56. Two periods.

Thank you for spotting this, one has now been removed. 

Materials and Methods

 6. The section 2.2.1 as it is, is not useful. Without specifying which test was considered and which parameters were used, those numbers have no meaning. I understand that the sample size calculation may have been conducted in the design phase when the planned statistical analyses were different from the ones conducted. Either provide a complete description of the rationale for the sample size calculation, or remove the text.

As the sample size calculation no longer relates to the analyses in the paper this section has now been removed (see page 5, lines 107-109). 

 7. Please modify Table 1 according to the following suggestions:

 o Rename the column “Description” as “Definition”

 o The definitions reported for sensitivity and specificity are incorrect. Please provide formal definitions. The text as it is, is very misleading. For example: 1) sensitivity is not “ability”, it is a probability. 2) sensitivity is not the probability of the test to correctly identify those who have motor difficulties…it’s the probability of being classified as having motor difficulties by the test GIVEN that the child actually has motor difficulties

 o Add formal definition also for PPV and NPV

 o Classification rate is not a metric. The metric described by the authors is called “Accuracy”. The definition of Accuracy should not be formulated as a question. Please add a formal definition.

 o Add a column, called “Operationalization”. In this column, explain how you calculated the quantities for each metric, exactly as you had done for PPV and NPV in the column currently called “Description”.

 o Remove the “(n=22)”. This is not the appropriate table to report sample sizes.

 o Change the title of the table in accordance with the name given to the analyses in the text

Thank you for this helpful comment - we have made the changes listed above, and agree they increase the clarity of the manuscript for those unfamiliar with these analyses. 

 Results

 8. Sample sizes are much more clear now. Please make the following edits to Table 2:

 - Rename “inter-rater reliability” with “Subsample used for inter-rater reliability analysis”. Following the same rationale, also rename “concurrent and predictive validity”.

 - In the first column, below “Sex”, there should be two cells. One cell with “Male, n(%)” and one with “Female, n(%)”. This way, you do not have to repeat in the other three columns “Male: n=” and “Female:=”. This is usually how sex is reported in Tables.

This has now been changed - thank you for this helpful comment, it has improved the readability of the table. 

 9. Section 3.2 Inter-rater reliability. There cannot be a gold standard of inter-rater reliability by definition. Please avoid the use of this term when referring to the inter-rater reliability assessment, as it is incompatible with the chosen statistical methods.

This has now been removed.

10. Table 3. Provide confidence intervals for the linear weighted Kappa. Please also delete the mention to “p<.001” in the caption, as no p-value is reported in the table.

 Apologies for the oversight, p<.001 has been removed from the footnote, and confidence intervals for the linear weighted kappa have now been included (see Table 3, pages 11-12). 

11. Table 4 should be a cross-table with the classification of the FUNMOVES and the one of MABC-2 total score.

We are unsure what is meant by this comment. We inferred this might be a suggestion that we should omit the Aiming & Catching and 

Balance subscale row. However we have decided to keep this row because it has now been clarified in the methods (see response to comment 13). 

12. The concurrent and predictive validity has not been defined as a sensitivity analysis in the methods. Please modify the language.

The table name and text above have now been changed to reflect the content (see page 12, lines 221-225). 

13. The MABC-2 Aiming and Catching and Balance subscales (FMS) has not been mentioned at all in the methods and it is unclear where it comes from.

This has now been included in the methods section (see page 9 lines 193-195). 

 Discussion

 14. When commenting on the results from the predictive validity analysis, please use correct definitions of the metrics (sensitivity, specificity, PPV, NPV). Do not oversimplify their meaning.The claim “the classification analyses were all strong” is unclear and is not scientific language. (1) Classification analysis is not a recognized term to refer to predictive validity analysis. (2) The phrase “all strong” is nondescript and I therefore don’t know what it means.

We thank the reviewer for highlighting the confusing nature of the discussion around sensitivity, specificity, PPV, NPV. We have edited this section to improve readability and understanding (see pages 14&15, lines 250-275). 

15. Page 14, line 278. Should say “could impact the generalizability of results”. Please use appropriate terms for scientific concepts.

The language in this sentence has now been clarified (see page 15, line 284). 

16. Page 14, line 283 to page 15 line 288. The fact that demographic factors affect the condition (fundamental movement skill ability levels) does not justify the correction by demographic factors for the test. It has been mathematically proven that correction for demographic factors can reduce the accuracy of tests under certain causal scenarios (this has been shown extensively for cognitive screening tests, for example). It is not “essential to formulate a normative database” until it is shown that the correction for demographic factors improves the metric of interest. This has not been demonstrated in this paper and therefore, it is not true that this “current study shows that such a programme of work is well justified”. In the next paragraph of the discussion, the authors continue to refer to the need for a normative database.

We appreciate the reviewer's opinion. The wording has been changed, rather than removing this section, because positive results (showing that FUNMOVES may be useful as a screening tool for identifying difficulties, in alignment with a widely used clinical tool) in a preliminary study such as those presented in this manuscript (where the limitations of this are made explicit) do justify further exploration, in larger sample sizes. See page 16, lines 291-295).

17. Line 326. “Psychometric” refers to psychological tests.

The language in this sentence has now been clarified (see page 17, line 332).

---

## [Decision Letter · Decision Letter 2]

6 Jun 2023

PONE-D-22-29655R2

The validity and reliability of school-based fundamental movement skills screening to identify children with motor difficulties

PLOS ONE

Dear Dr. Eddy,

Thank you for submitting your manuscript to PLOS ONE. After careful consideration, we have decided that your manuscript does not meet our criteria for publication and must therefore be rejected.

I am sorry that we cannot be more positive on this occasion, but hope that you appreciate the reasons for this decision.

Kind regards,

Gorica Maric

Academic Editor

PLOS ONE

Additional Editor Comments:

Dear authors,

Although your manuscript has undergone two rounds of review process, one of the reviewers is still not satisfied with your corrections especially in terms of clarification of terminology used in the manuscript as well as statistical analysis. Keeping in mind importance of these concepts when performing validation study we couldn't accept the manuscript at the moment. At the bottom of this page you can find some additional comments that can further improve your work.

Reviewers' comments:

Reviewer's Responses to Questions

**Comments to the Author**

1. If the authors have adequately addressed your comments raised in a previous round of review and you feel that this manuscript is now acceptable for publication, you may indicate that here to bypass the “Comments to the Author” section, enter your conflict of interest statement in the “Confidential to Editor” section, and submit your "Accept" recommendation.

Reviewer #1: (No Response)

Reviewer #2: All comments have been addressed

2. Is the manuscript technically sound, and do the data support the conclusions?

Reviewer #1: No

Reviewer #2: Yes

3. Has the statistical analysis been performed appropriately and rigorously? 

Reviewer #1: I Don't Know

Reviewer #2: Yes

4. Have the authors made all data underlying the findings in their manuscript fully available?

Reviewer #1: Yes

Reviewer #2: Yes

5. Is the manuscript presented in an intelligible fashion and written in standard English?

Reviewer #1: Yes

Reviewer #2: Yes

6. Review Comments to the Author

Reviewer #1: - FMS acronym is not defined. I suggest to avoid this acronym and spell the terms out.

- Page 8, lines 179-180. RStudio runs using R, please, specify which underlying version of R was used for the analyses

- Table 1 still contains problematic statements.

o Operationalization for “Sensitivity” is wrong. Sensitivity is not calculated as “What is the probability that FUNMOVES is correctly classifying children that have motor difficulties (scoring <15th percentile on the MABC)”, it should be “What is the probability that FUNMOVES classifies children as having motor difficulties ([explain what’s the threshold used for FUNMOVES]) given that the children have motor difficulties (they score <15th percentile on the MABC)”.

o Same issue applies to Specificity.

o The definition of PPV is factually wrong. PPV is the probability that a child actually has motor difficulties given that the test classifies the child as having motor difficulties. The operationalization should be: PPV is the probability that children have motor difficulties (they score <15th percentile on the MABC) given that they were classified as having motor difficulties by the FUNMOVES ([explain what’s the threshold used for FUNMOVES]).

o Same applies to NPV

o Accuracy is the probability that a child is correctly classified by FUNMOVES. The operationalization should be “What is the probability that FUNMOVES classifies children in the same way as MABC?”

o All the formulas are wrong,since parentheses are not used. For example, sensitivity is obviously not “TP/TP+FN” but “TP/(TP+FN)”.

- Again, remove “gold standard” from table 3. MABC can be considered a gold standard, but the FUNMOVES assessment of the researcher is not a gold standard.

- In table 3 the Balance linear Kappa does not have 95% CI

- Page 12, lines 228-234. These analyses have not been explained nor justified. They do not seem to be pre-specified analysis and therefore are questionable.

- The sentence “Analyses revealed that the probability of FUNMOVES correctly classified all children that do not have motor difficulties” is wrong. This is not the definition of specificity.

- “Additionally, FUNMOVES had a positive predictive value of 1 (95%CI =0.68-1) such that the probability that children who score below the 15th percentile also score below this threshold on the MABC-2 total score is high.” This sentence is also wrong. Please fix it.

- The same applies to the discussion of Sensitivity and NPV.

- Page 15, line 276. Accuracy is a probability exactly as the other measures, use the same scale.

- Half of the discussion seems to focus on the need of a normative dataset for FUNMOVES to adjust for age. This is not justified by the current work, nor does not seem relevant to the current paper.

Reviewer #2: (No Response)

7. PLOS authors have the option to publish the peer review history of their article (what does this mean?). If published, this will include your full peer review and any attached files.

Reviewer #1: No

Reviewer #2: No

- - - - -

---

## [Author Response · Author response to Decision Letter 2]

14 Jul 2023

Rebuttal of reviewer comments

1. Table 1 still contains problematic statements….

o Operationalization for “Sensitivity” is wrong. Sensitivity is not calculated as “What is the probability that FUNMOVES is correctly classifying children that have motor difficulties (scoring <15th percentile on the MABC)”, it should be “What is the probability that FUNMOVES classifies children as having motor difficulties ([explain what’s the threshold used for FUNMOVES]) given that the children have motor difficulties (they score <15th percentile on the MABC)”.

o Same issue applies to Specificity.

o The definition of PPV is factually wrong. PPV is the probability that a child actually has motor difficulties given that the test classifies the child as having motor difficulties. The operationalization should be: PPV is the probability that children have motor difficulties (they score <15th percentile on the MABC) given that they were classified as having motor difficulties by the FUNMOVES ([explain what’s the threshold used for FUNMOVES]).

o Same applies to NPV

o Accuracy is the probability that a child is correctly classified by FUNMOVES. The operationalization should be “What is the probability that FUNMOVES classifies children in the same way as MABC?”

o All the formulas are wrong,since parentheses are not used. For example, sensitivity is obviously not “TP/TP+FN” but “TP/(TP+FN)”.

o The sentence “Analyses revealed that the probability of FUNMOVES correctly classified all children that do not have motor difficulties” is wrong. This is not the definition of specificity.

o “Additionally, FUNMOVES had a positive predictive value of 1 (95%CI =0.68-1) such that the probability that children who score below the 15th percentile also score below this threshold on the MABC-2 total score is high.” This sentence is also wrong. Please fix it.

o The same applies to the discussion of Sensitivity and NPV.

We amended these definitions and operationalisations in alignment with the reviewer’s comments from last time. To avoid confusion, we used the exact terminology from their reviews. They are now suggesting that the wording they recommended is inappropriate and 'problematic'. Inconsistencies like this raise concerns that this reviewer is being deliberately obstructive. Papers have been published and highly cited within Developmental Psychology with similar definitions (e.g. Atkinson et al. 2022) which guided the ‘operationalisation’ column in this paper. The reviewer is ultimately arguing over semantics. We have changed these to standardised cited definitions to avoid any further confusion, and aligned the operationalisation of these to Atkinson et al. (2022). We have also included parentheses in the formulas. 

2. Page 12, lines 228-234. These analyses have not been explained nor justified. They do not seem to be pre-specified analysis and therefore are questionable.

We responded to the expert advice provided by Reviewer 1 in the first round of revisions to remove regressions and correlations from the paper and refocus the analysis around sensitivity, specificity etc (which were included in the original manuscript). In order to undertake these analyses, true positives, true negatives, false positives and false negatives need to be calculated beforehand. These are standard metrics that other papers looking to test agreement use regularly (e.g. Atkinson et al., 2022). Furthermore, their criticism of our lack of pre-registration is extremely unreasonable. We can't make retrospective changes to the analysis used here, which were expressly requested by this reviewer and travel back in time to pre-register them! This is a criticism that has only arisen because we have been responsive to their early criticisms of our analysis and tried to address these, post-hoc.

3. Half of the discussion seems to focus on the need of a normative dataset for FUNMOVES to adjust for age. This is not justified by the current work, nor does not seem relevant to the current paper.

There are four lines included in the discussion (out of 82 total), outlining the need to understand the applicability of this assessment tool beyond Bradford, in a more generalisable sample. Adjusting for age is not mentioned in these four lines as previous papers e.g. Eddy et al., 2021 (published in PLoS One) found that FUNMOVES does not have item response bias for age and thus there is no need to adjust for age. A normative dataset needs to be created which is nationally representative to gain more accurate percentile ranks for children’s ability levels. We fail to understand how the normal scientific process of generating and reporting positive, small-scale results as a precursor to a large-scale study can be criticised as being “unjustified”. 

4. Again, remove “gold standard” from table 3. MABC can be considered a gold standard, but the FUNMOVES assessment of the researcher is not a gold standard.

This is a misunderstanding from the reviewer. Gold standard scoring relates to the fact that the researchers involved in the development of the tool have the greatest knowledge on implementation of the tool. We have removed this to avoid confusion. 

5. FMS acronym is not defined. I suggest to avoid this acronym and spell the terms out.

We have removed acronyms throughout the manuscript.

6. Page 8, lines 179-180. RStudio runs using R, please, specify which underlying version of R was used for the analyses

This has now been clarified in the manuscript. 

7. In table 3 the Balance linear Kappa does not have 95% CI

The authors apologise for this oversight. This value has now been included in Table 3.

---

## [Decision Letter · Decision Letter 3]

31 Aug 2023

PONE-D-22-29655R3The validity and reliability of school-based fundamental movement skills screening to identify children with motor difficultiesPLOS ONE

Dear Dr. Eddy,

Thank you for submitting your manuscript to PLOS ONE. After careful consideration, we feel that it has merit but does not fully meet PLOS ONE’s publication criteria as it currently stands. Therefore, we invite you to submit a revised version of the manuscript that addresses the points raised during the review process.

We look forward to receiving your revised manuscript.

Kind regards,

Gorica Maric

Academic Editor

PLOS ONE

Journal Requirements:

2. Please expand the acronym “ESRC” (as indicated in your financial disclosure) so that it states the name of your funders in full.

Additional Editor Comments (if provided):

Reviewers' comments:

Reviewer's Responses to Questions

**Comments to the Author**

1. If the authors have adequately addressed your comments raised in a previous round of review and you feel that this manuscript is now acceptable for publication, you may indicate that here to bypass the “Comments to the Author” section, enter your conflict of interest statement in the “Confidential to Editor” section, and submit your "Accept" recommendation.

Reviewer #3: All comments have been addressed

Reviewer #4: (No Response)

2. Is the manuscript technically sound, and do the data support the conclusions?

Reviewer #3: Partly

Reviewer #4: Yes

3. Has the statistical analysis been performed appropriately and rigorously? 

Reviewer #3: I Don't Know

Reviewer #4: Yes

4. Have the authors made all data underlying the findings in their manuscript fully available?

Reviewer #3: No

Reviewer #4: No

5. Is the manuscript presented in an intelligible fashion and written in standard English?

Reviewer #3: Yes

Reviewer #4: Yes

6. Review Comments to the Author

Reviewer #3: The overall premise of the study is good and the reader is able to get the point that the author is trying to make. It is very difficult to understand the numbers and the calculations. A flowchart that includes the numbers in different stages would be helpful for organization purposes and to make the paper easier to read. There are different numbers of participants at the beginning 217 and at the end (22). A flowchart that explains the exclusion of the subjects starting from the top to the end would be helpful. The same applies with to the teachers and researchers. Did the same teacher and researcher pair evaluate different students? The authors present a table with definitions of sensitivity, specificity, PPV, NPV. In addition to the explanation of these terms, they should put the numbers that applies to their study in that table.

Reviewer #4: There is improvement in the quality of the manuscript with the adjustment made based on reviewers' suggestions.

There is need to update the following information:

1. Abstract: FMS to ne written in full.

2. Introduction: The authors should state in a sentence the difficulties with using MABC especially in school setting.

3. Methods: There is need to input how the researchers assessed each teacher's competence in using the tool after an hour training.

The sampling method (convenient)should be stated as part of the limitation to this study3. Procedure: Line 148: skills not sills

Line 92: The school was and not were.

4. Line 167, 200: The school identified 30 children should be changed to the schoolteachers.

5. Line 201: 22 completed FUNMOVES due to school absenteeism-This information is not clear. This study was done in a day and no drop out from parental consent. Authors need to clarify the information about school absenteeism.

6 There is need to provide some information on the FUNMOVES such as, has it been used in previous study, was there any study that has don similar study,

7. Discussion: This seems to be narrowed. There is need for the authors to compare their findings (results) with previous studies).

7. PLOS authors have the option to publish the peer review history of their article (what does this mean?). If published, this will include your full peer review and any attached files.

Reviewer #3: No

Reviewer #4: No

---

## [Author Response · Author response to Decision Letter 3]

6 Nov 2023

Rebuttal of reviewer comments

Reviewer 3 

1. The overall premise of the study is good and the reader is able to get the point that the author is trying to make.

We thank the reviewer for their kind words.

2. It is very difficult to understand the numbers and the calculations. A flowchart that includes the numbers in different stages would be helpful for organization purposes and to make the paper easier to read. There are different numbers of participants at the beginning 217 and at the end (22). A flowchart that explains the exclusion of the subjects starting from the top to the end would be helpful. The same applies with to the teachers and researchers. Did the same teacher and researcher pair evaluate different students? 

A flow chart was deemed excessive as there were no exclusions and no non-consents. It would only illustrate the number of children on the school roll, as all were invited to participate, flowing to a second box showing the number of children who did participate. Teachers were asked to identify children that they believed were struggling from the total sample (concurrent and predictive) – this has been clarified in the manuscript (see page 10, lines 204-210). In order to test whether teachers and researchers scored the same children in the same way (inter-rater reliability) they had to score the same students. 

3. The authors present a table with definitions of sensitivity, specificity, PPV, NPV. In addition to the explanation of these terms, they should put the numbers that applies to their study in that table.

This table is within the methods section to explain the definitions to a lay reader. The results are presented in a separate table. We have included reference to the definitions table in as an NB in the footer, to refer readers back to definitions. 

Reviewer 4 

4. Abstract: FMS to ne written in full.

We thank the reviewer for highlighting an omission in defining an acronym. We have been through the abstract (and the rest of the manuscript) and removed all acronyms (except the MABC-2 as this is a well-known acronym for a clinically used tool). 

5. Introduction: The authors should state in a sentence the difficulties with using MABC especially in school setting.

We agree that the specific difficulties of implementing the MABC being more explicit within the manuscript would be beneficial. We have now included a sentence about this in the introduction (see page 4, lines 74-75).

6. Methods: There is need to input how the researchers assessed each teacher's competence in using the tool after an hour training.

Implementation fidelity was not formally assessed, as extensive work was done on this during the development of the tool. A line about this has now been included in the manuscript (see page 7, lines 163-165).

7. The sampling method (convenient) should be stated as part of the limitation to this study

We agree this is a useful addition – and ties into discussions around generalisability already present within the discussion. We have made it explicit in the discussion that convenience sampling may also contribute to this (see page 15, lines 289-290).

8. Procedure: Line 148: skills not sills

We thank the reviewer for spotting this error – we have now rectified this within the manuscript (see page 7, line 152)

9. Line 92: The school was and not were.

We thank the reviewer for highlighting this error – we have now changed this within the manuscript (see page 5, line 98)

10. Line 167, 200: The school identified 30 children should be changed to the schoolteachers.

We thank the reviewer for highlighting this discrepancy – we have now changed this within the manuscript (see page 8, line 173)

11. Line 201: 22 completed FUNMOVES due to school absenteeism-This information is not clear. This study was done in a day and no drop out from parental consent. Authors need to clarify the information about school absenteeism.

We apologise for the ambiguity in this section – we have changed the sentence to incorporate more clarity (see page 10, lines 211-212).

12. There is need to provide some information on the FUNMOVES such as, has it been used in previous study, was there any study that has don similar study. 

Discussion: This seems to be narrowed. There is need for the authors to compare their findings (results) with previous studies).

The first study looking at the inter-rater reliability, and concurrent and predictive validity of FUNMOVES. Only one previous published study involves the assessment tool (as it is newly developed, and has not been made readily available until psychometric properties are established). It is therefore difficult to make comparisons to other studies, as there are no other relevant publications. We have included a sentence in the methods referring readers to the first FUNMOVES paper – outlining its development and what it incorporates (see page 6, lines 129-130). We have also clarified in the discussion that this is the first study to look at these aspects of validity and reliability (see page 14, line 254).

---

## [Editor Report · Decision Letter 4]

4 Jan 2024

The validity and reliability of school-based fundamental movement skills screening to identify children with motor difficulties

PONE-D-22-29655R4

Dear Dr. Eddy,

We’re pleased to inform you that your manuscript has been judged scientifically suitable for publication and will be formally accepted for publication once it meets all outstanding technical requirements.

Kind regards,

Gorica Maric

Academic Editor

PLOS ONE
---

## [Editor Report · Acceptance letter]

7 Feb 2024

PONE-D-22-29655R4 

PLOS ONE

Dear Dr. Eddy, 

I'm pleased to inform you that your manuscript has been deemed suitable for publication in PLOS ONE. Congratulations! Your manuscript is now being handed over to our production team.

Kind regards, 

on behalf of

Dr. Gorica Maric 

Academic Editor

PLOS ONE